# Cross-View Outdoor Localization in Augmented Reality by Fusing Map and Satellite Data

René Emmaneel [1,2,*], Martin R. Oswald [1], Sjoerd de Haan [3] and Dragos Datcu [2,*]

1    Computer Vision Group, Informatics Institute, Faculty of Science, University of Amsterdam, Science Park 904, 1098 XH Amsterdam, The Netherlands
2    Huawei Technologies Netherlands, 1101 CM Amsterdam, The Netherlands
3    Go Grow AI, 1076 VC Amsterdam, The Netherlands
*    Correspondence: rene.emmaneel00@gmail.com (R.E.); d.datcu@huawei.com (D.D.)

**Abstract:** Visual positioning is the task of finding the location of a given image and is necessary for augmented reality applications. Traditional algorithms solve this problem by matching against premade 3D point clouds or panoramic images. Recently, more attention has been given to models that match the ground-level image with overhead imagery. In this paper, we introduce AlignNet, which builds upon previous work to bridge the gap between ground-level and top-level images. By making multiple key insights, we push the model results to achieve up to 4 times higher recall rates on a visual position dataset. We use a fusion of both satellite and map data from OpenStreetMap for this matching by extending the previously available satellite database with corresponding map data. The model pushes the input images through a two-branch U-Net and is able to make matches using a geometric projection module to map the top-level image to the ground-level domain at a given position. By calculating the difference between the projection and ground-level image in a differentiable fashion, we can use a Levenberg–Marquardt (LM) module to iteratively align the estimated position towards the ground-truth position. This sample-wise optimization strategy allows the model to align the position better than if the model has to obtain the location in a single step. We provide key insights into the model's behavior, which allows us to increase the model's ability to obtain competitive results on the KITTI cross-view dataset. We compare our obtained results with the state of the art and obtain new best results on 3 of the 9 categories we look at, which include a 57% likelihood of lateral localization within 1 m in a 40 m × 40 m area and a 93% azimuth localization within 3° when using a 20° rotation noise prior.

**Keywords:** augmented reality; visual positioning; outdoor localization; maps; satellite data

## 1. Introduction

Finding an image's exact location and orientation is a challenging task, but it is essential to obtain these data accurately for the purpose of augmented reality applications where an image's precise location is important, such as in AR navigation. Using GPS data, an image's location can be localized to a small area; however, the precision of this approach is limited. This problem is an active topic of research, and previous attempts have included using ground-level images for 3D model mappings [1–4] and ground-level images for 2D street view images mapping [5,6]. These techniques, however, all share a common problem: the data to map against are hard to acquire. Creating a 3D model of a city is tough, and although many techniques to build 3D models of cities exist [7,8], making a robust model that does not rely on these models would allow for scaling beyond preprocessed areas. Otherwise, street view data can be outdated in just a few years or when the seasons change [9].

To circumvent these issues, we propose a technique based on top-down imagery, such as satellite and OpenStreetMap (OSM) data, instead. This technique has been an active topic

of research for the last few years [10–19]. The basis of these techniques involves matching the ground-level image to specific positions in the top-level image to estimate where the image was taken. In this paper, we build upon previous work to build a high-performing model for this task.

The task is a particularly difficult challenge because of the large information gap between the top-level and ground-level images. Previous work used metric learning to learn deep representations of both modalities to make matches without taking the geometry into account [19]. To go beyond metric learning, we base our model on the work by Shi et al., who used geometric projection to bridge this gap [13]. In particular, we use a U-Net architecture for both top-level and ground-level input, use a geometric projection module to translate the top-level input to the ground-level modality, and use the Levenberg–Marquadt optimization algorithm [20] to optimize the position over steps. The LM algorithm aims to align the projected top-level image with the ground-level image. As such, we call our model AlignNet, of which we give an overview in Figure 1.

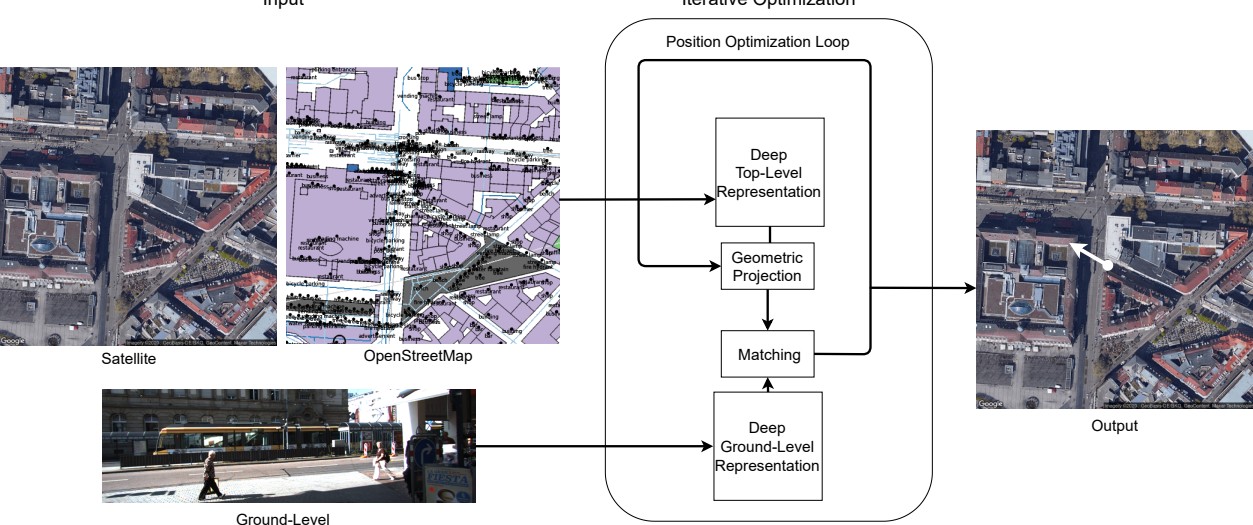

**Figure 1. Model overview.** Input consists of top-level and ground-level imagery. AlignNet matches the projected top-level image with the ground-level image and iteratively aligns towards a single output position. The output consists of a lateral, longitudinal, and angular position.

To go beyond the previous work, we make significant contributions to their model and show the benefits of those contributions. In particular, we introduce a fusion of satellite and OpenStreetMap data to improve the input; we also improve the architecture using a different U-Net architecture and show improvements regarding how the model is trained. Furthermore, we contribute towards the research of this iterative sample-wise optimization model by providing an analysis of the behavior of this model. Finally, we conduct an ablation study on the model to show the importance of the various design choices made in our model.

The next section discusses recent works on visual localization using satellite data. In Section 3, we describe our augmented reality-related method for cross-view outdoor localization by fusing map and satellite data. Then, in Section 4 we analyze the performance of the model proposed and also provide information regarding an ablation study. In the last section, we discuss and conclude on the findings of the current research.

## 2. Related Work

The related work concerning visual localization using satellite data can be separated into three different branches: visual localization without deep learning, global visual localization with deep learning, and local visual localization with deep learning. We first

summarize the three different branches and then look at work related to creating datasets for these tasks.

### 2.1. Localization without Deep Learning

Early works on this topic made various attempts at creating models without using deep features. Castaldo et al. made use of segmentation models and horizon estimation to transform ground-level images to a rectified view to match the top-down view [21]. Lin et al. matched ground-level images with other similar images to average over the features and used that as input to an SVM classifier to locate the image [22]. Mousavian and Kosecka used the geometric information of facades in ground-level images to create a probabilistic location map based on a map of the area [23].

### 2.2. Large-Scale Localization with Deep Learning

Various attempts at this taskhave been made based on the success of deep learning in various computer vision tasks. Earlier works in this field mostly looked at matching ground-level images with satelite images and were mostly formulated as retrieval tasks. Workman and Jacobs finetuned a CNN architecture trained on imagenet to match ground-level images and satellite images [24].

Models utilizing Siamese networks such that the ground-level and top-down images share the same models, without weight sharing, have been researched [19,25–29]. These models are typically learned using a triplet loss [30], which is a loss function that compares an input to a positive and negative example and minimizes the distance to the positive example, while the distance to the negative example is maximized.

### 2.3. Small-Scale Localization with Deep Learning

Recently, work has been conducted to find an image's precise pose on a single satellite image. Vigor [19] simultaneously optimized for the satellite image retrieval task as well as the pose estimation task. Other research split satellite data into patches and used cosine similarity to match a deep representation of the ground image; this was then used to obtain a probability density map for the localization [17]. In a follow-up work, they introduced rolling descriptor matching and created a probability density map over multiple levels to refine their estimations [16]. Research has been also been conducted using transformers with the aim of achieving results beyond those obtainable by CNN-based models [18].

Furthermore, the localization estimate can be updated over time by utilizing particle filters [25,26].

To close the large modality gap between ground-based images and satellite images, Shi et al. used polar-transformed aerial imagery to match both cropped and uncropped 360 degree panoramic images [14]. In a follow-up work, Shi et al. introduced a model that utilized a U-net-based architecture for both the ground-level image and satellite image [13]. They used a projection step with estimated position and orientation and calculated the mean square error of the projection and deep representation of the ground-level image. Their model optimized the estimated pose using the Levenberg–Marquardt optimization algorithm within several iterations [20]. Their network was fully differentiable, which allowed the model to backpropagate the pose loss over the weights of the feature extractors.

Afterwards, SliceMatch [11] was proposed, which used masking to efficiently calculate the cosine similarity between the ground-level image and many different poses on the masked satellite image.

Other methods have included the use of multiple ground-level images and a perspective view to bird's eye view transformation to create uncertainty-aware visual localization [10] and have used LIDAR for better pose estimation [15].

The current best approach, however, comes from utilizing OpenStreetMap data instead of satellite data [31]. These data include rich tagged information and are noticeably less noisy. By extracting node, line, and area data as segmented data and using a larger

training dataset, OrienterNet models a probability map that outperforms satellite-only approaches [12].

## 3. Methods

In this section, we provide a detailed description of the proposed method. First, we discuss the details of the dataset and how the top-down images are extracted, including the OpenStreetMap data. Afterwards, we discuss the details of the model.

### 3.1. Data Overview

The input data of the model were the KITTI dataset [32], as well as corresponding satellite and OSM data. The satellite data represent a 250 m by 250 m area around the ground-truth position and have a resolution of 1280 pixels by 1280 pixels, as extracted by Shi et al. [13]. Furthermore, we extracted OSM data from the exact same position and area by utilizing the overpass API and mapping each node, line, and area to a specific semantic class.

Before the input data were put into the encoders, translation and rotation noise were added to the image. The center 100 m by 100 m area, corresponding to an image of 512 by 512 pixels, was cut out. This area was made larger than the maximum translation noise to ensure the model worked properly even at the border of the translation noise. For the OSM data, the same preprocessing steps were taken to ensure the satellite and OSM data were exactly aligned.

#### OpenStreetMap Extraction

Following the work by Sarlin et al. [12], we, instead of directly taking a screenshot-like image of OpenStreetMap, extracted its node, line, and area (closed-loop line objects) objects. Based on the tags included in these objects, we manually constructed rules to map useful objects to different classes. These classes included areas such as buildings or vegetation; the lines included different types of roads, and nodes included many different single-place positions including trees, doors, and signs. Although later, we show that the main contributing modality to the improvement seen when using the OSM data is the road data, we used all 3 modalities as input to the model. This was carried out by setting each of the 3 channels to a certain value based on the geometry of the input OSM objects and for each pixel of the OSM input (Figure 2). The resolution was 512 pixels by 512 pixels, which corresponded to about $\Delta$50 cm per pixel, with lines and nodes having a width of 4 pixels.

### 3.2. Model Overview

The model is based on the original model introduced by Shi et al. [13]. That model used a geometric projection to match BEV (birds-eye view) and ground-level features and iteratively optimized the estimation process over a number of iterations. The BEV input contained a satellite image of the area and optionally augmented these data with the OpenStreetMap data for the same area [31]. A We split the model up into four parts: (1) feature extractors for both the input modalities, (2) geometric map matching, (3) a Levenberg–Marquardt optimization algorithm, and (4) a loss function. An overview of this is shown in Figure 3.

#### Feature Extractors

First, the deep features of the BEV and ground-level images were extracted using a U-Net architecture [33] that was initialized using the pretrained VGG13 [34] weights with batch normalization in the encoding layers. This is a bigger and more conventional U-Net architecture than the architecture Shi et al. uses, which we observed to improve results.

### 3.3. Top-Level to Ground-Level Projection

We provide details of our geometric projection, which was carried out following the steps for implementation provided by Shi et al. [13]. The geometric projection module projects the deep-encoded top-level images to the ground-level. This was carried out

using a pinhole geometric projection from a point on the top-level image, where we used a flat-world assumption. The camera was set a fixed amount above this flat plane and with a given camera intrinsic *K*, as given by the KITTI dataset.

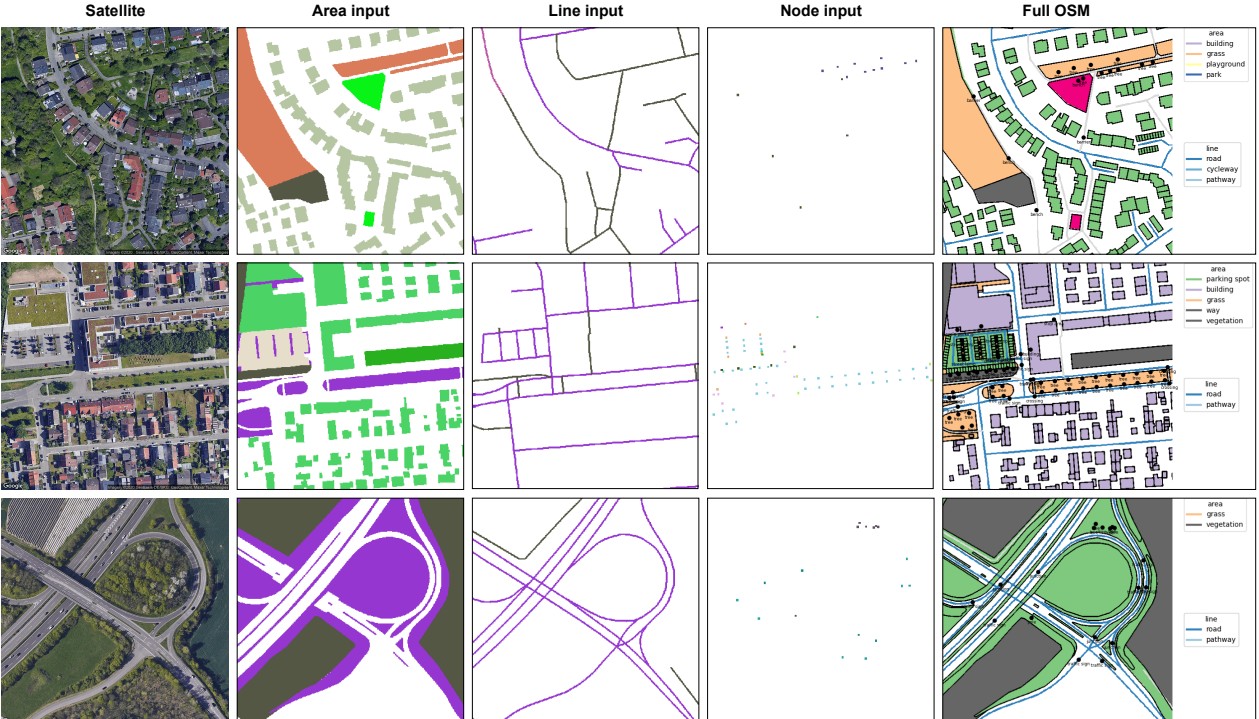

**Figure 2. OpenStreetMap input examples.** Left columns shows pre-cropped satellite image. The right-most column shows the data, which were extracted with a legend for human readability, while columns 2–4 shows the input the model receives. The extracted images are $512 \times 512$ pixels, with roads and nodes having a set size of 4 pixels and where each different color is a different class. The outlines of each areas as shown in the full OSM column are not part of the input to the model.

The world coordinate system was set to the center of the top-level image; the *x* axis was parallel to the $v^s$ direction of the top-level image, the *y* axis was pointing downward, and the *z* axis was parallel to the $u^s$ direction. A 3D point $(x, y, z)$ in the world coordinate system was mapped to a top-level image pixel using the orthogonal projection,

$$\left[ \; u_s, v_s \; \right] = \left[ \; \frac{z}{\alpha} + u_s^0, \frac{x}{\alpha} + v_s^0 \; \right]^T, \tag{1}$$

where $\alpha$ is the meter-per-pixel distance of the top-level image and $(u_s^0, v_s^0)$ correspond to the top-level image's center.

Supposing a rotation given by matrix **R** and translation given by vector **t**, we calculate the world coordinates as follows:

$$\left[ \; x, y, z \; \right]^T = \mathbf{R}(\left[ \; x_c, y_c, z_c \; \right]^T + \mathbf{t}). \tag{2}$$

The projection from the 3D point to the pin-hole camera image plane is formulated as

$$w \left[ \; u^g, v^g, 1 \; \right]^T = \mathbf{K} \left[ \; x_c, y_c, z_c \; \right]^T, \tag{3}$$

where **K** is the camara intrinsic and *w* is a scale factor.

Following Equations (1)–(3), Shi et al. derived the following mapping from a ground-view pixel to a satellite pixel as

$$\begin{bmatrix} u_s \\ v_s \\ z \end{bmatrix} = \begin{bmatrix} \frac{1}{a} & 0 & 0 \\ 0 & \frac{1}{a} & 0 \\ 0 & 0 & 1 \end{bmatrix} \left( w\mathbf{R}\mathbf{K}^{-1} \begin{bmatrix} u_g \\ v_g \\ 1 \end{bmatrix} + \mathbf{R}\mathbf{t} \right) + \begin{bmatrix} u_s^0 \\ v_s^0 \\ 0 \end{bmatrix}. \tag{4}$$

As said before, we make a flat-world assumption in these calculations. Given these equations, this is easy to do by setting $y_c$ in Equation (3) to the distance from the camera to the ground plane. As the camera height is given, this is easy to calculate. Then, the scale factor $w$ can be calculated using that equation. Alternatively, if this distance is not known such as in a diverse dataset, a strategy of using multiple different scale factors could be used [12].

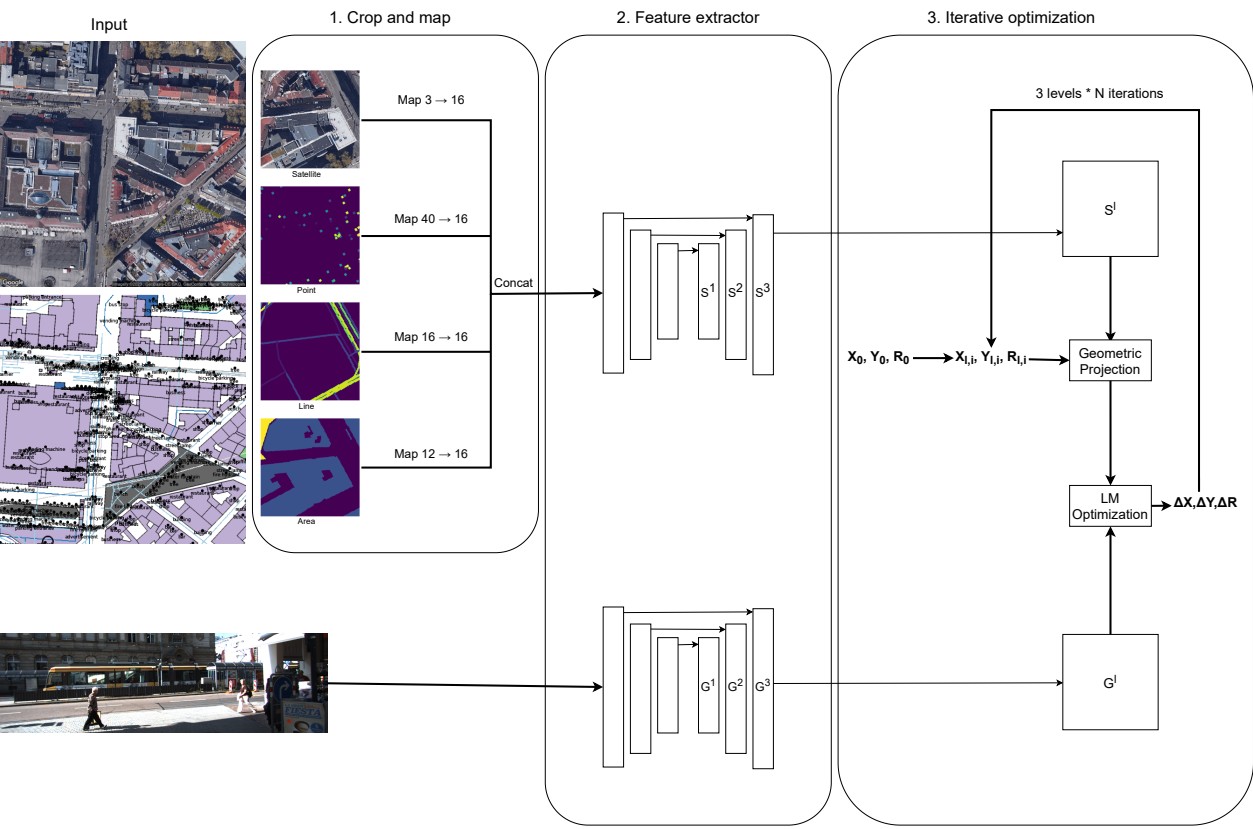

**Figure 3. Overview of AlignNet.** (1) The top-down input consists of satellite and OpenStreetMap data, which are then cropped and rotated around the ground-truth point with the true heading facing to the right and with randomly added translation and rotation noise. The 4 modalities are then mapped through a learned mapping and concatenated. (2) The top-down and street-level inputs are put through a two-branch U-Net feature extractor, resulting in encoded features at different levels called $S^1, \ldots, S^3$ and $G^1, \ldots, G^3$, respectively. (3) Next, starting at position $\boldsymbol{X_0}, \boldsymbol{Y_0}, \boldsymbol{R_0}$, the model maps the features $S^l$ to the ground-level domain. It tries to minimize the difference between the mapping and $G^l$ by optimizing the pose variables using LM optimization, going first through each level, which is then repeated $N$ times.

### 3.4. Levenberg–Marquardt Optimization

To optimize the positional variables, we used the optimizer implemented by Shi et al. [13], which aimed to minimize the difference between the top-level and ground-level. After

calculating the deep features from the input images, we denoted the difference of the projected top-level image and ground-level image as

$$\mathbf{e^l} = \mathbf{P^l} - \mathbf{G^l}, \tag{5}$$

where $\mathbf{P^l}$ is the projected top-level features at level $l$ and $\mathbf{G^l}$ is the ground-level features at level $l$. We minimize this difference by optimizing the positional variables $\xi = \{\mathbf{R}, \mathbf{t}\}$. Our objective becomes

$$\hat{\xi} = \underset{\xi}{\mathrm{argmin}} ||\mathbf{e^l}||_2, \tag{6}$$

where $||.||_2$ denotes the L2 norm and $\hat{\xi}$ are the optimizal positional variables. Given the small number of variables to be optimized, we can solve this using the Levenberg–Marquardt (LM) optimization algorithm [20].

For each level $l$, we calculated the Jacobian matrix and Hessian matrix with regards to the positional variables $\xi$,

$$\mathbf{J} = \frac{\partial \mathbf{P^l}}{\partial \xi} = \frac{\partial \mathbf{P^l}}{\partial \mathbf{p}_s} \frac{\partial \mathbf{p}_s}{\partial \xi}, \quad \text{and} \quad \mathbf{H} = \mathbf{J}^T \mathbf{J}, \tag{7}$$

where $\mathbf{p_s}$ is the satellite's feature map coordinates. We followed the original implementation and used Levenberg's damping formula to interpolate between gradient descent (at $\lambda = \infty$) and Gaussian–Newton (at $\lambda = 0$), as formulated by

$$\tilde{\mathbf{H}} = \mathbf{H} + \lambda \mathbf{I}. \tag{8}$$

We used the same value of $\lambda = 0.1$ as in prior work [13]. After calculating $\tilde{\mathbf{H}}$, we can update the pose by

$$\xi_{t+1} = \xi_t + \tilde{\mathbf{H}}^{-1} \mathbf{J}^T \mathbf{e}, \tag{9}$$

where $t$ denotes the current step.

The LM optimization was first applied to the coarsest feature level before going to the finer levels, for a total of 3 levels. This repeats 5 times before the model is finished, and the final position is compared to the ground-truth position. This scheme was chosen with the idea of the coarser levels being able to escape local minima while the finer levels can be used to align the final position at a finer scale. We found that theory did not hold up well in practice, although different levels were more well-suited to updating certain positional variables.

Because this model is end-to-end differentiable, the positional loss can be backpropagated to the feature encoders. Furthermore, the loss between the encoded ground-level features and encoded and top-down features projected at the ground-truth position can be calculated using an L2 loss directly and can be optimized directly to stabilize the training process. This must be carried out carefully as the model could collapse an project all input pixels to the same value, which would result in a zero-valued L2 loss.

### 3.5. Starting Position

A perhaps unremarkable but important design decision for the model is the initial guess of the model. We denote the different strategies we used and provide a small justification for our decision before delving deeper into the nuances in the Experiments section. The easiest strategy and the strategy followed by Shi et al. is to start with all positional variables set to 0. As such, we will call this strategy *center*. One important aspect of how the model works is that during the projection step, the model projects only to pixels in front of it, and as such, it is unable to look backwards. Because of that, an improved strategy is to start at the minimum possible longitudinal value, which we will simply denote as *lon - 1*. A summary of the starting position strategies we used is given in Table 1. Further strategies that were explored but that did not show consistent improvements included

either randomizing certain or all positional variables and training the model to guess an initial position close to the ground truth before starting the LM optimization process.

**Table 1. Summary of the different starting strategies used.** Choosing how the positional variables are initialized before the LM optimization has a large and explainable effect on the final results. $-1$ refers to the minimum possible value for that specific positional variable, while 1 refers to the maximum value.

| Name | Summary |
|---|---|
| center | Start with all positional variables set to 0 |
| *lon - 1* | Longitudinal is set to $-1$, rest to 0 |

*3.6. Loss Functions*

The conventional loss function we used is based on the positional loss at each step of the iteration. This is the obvious choice as the model is fully differentiable and so encoding the direct objective into the loss function helps the model perform best. However, due to the nature of the LM optimization algorithm, a loss function that minimizes the difference was considered as well.

As the entire network was end-to-end differentiable, we could directly backpropagate the positional loss to the feature encoders. Furthermore, we could add the intermediate position after each iteration to stabilize the training process. We denoted the following positional losses:

$$\mathcal{L}_{pos} = \sum_i \sum_l (|\hat{X}_i^l - X^*| + |\hat{Y}_i^l - Y^*| + |\hat{R}_i^l - R^*|), \tag{10}$$

where $\hat{X}_i^l$, $\hat{Y}_i^l$ and $\hat{R}_i^l$ denote the predicted longitude, latitude, and angle at iteration $i$ and level $l$, while $X^*$, $Y^*$ and $R^*$ denote the GT pose.

As another type of loss function, we can calculate the L2 difference between the projection of the deep features at the ground-truth position and the deep features of the ground-level image by normalizing the features and calculating the sum of the element-wise square of the differences:

$$\mathcal{L}_{L2} = |(norm(proj(T)) - norm(G))^2|_1 \tag{11}$$

It is important to note that the model could optimize this loss by mapping all pixels to the same deep feature, resulting in a nondescript feature space. As such, instead of relying on this loss function, we only used it as a regularization term during the first epoch as a warm-up term. We decreased the importance linearly towards 0 during the first epoch, and afterward, we did not use this loss function anymore.

## 4. Results

In this section, we introduce the setup of the experiments and the evaluation metrics. We analyze our model by looking at the behavior of the model. We explain the motivation behind our design choices by providing an ablation study and show our contribution to the literature.

*4.1. Setup*

Our models were all trained for 21 epochs using a Tesla V100 GPU; most models tend to be converged at this point. The batch size was either 3 or 4 depending on the model due to the large amount of memory needed for storing the backpropagation graph as the graph increased in size over the LM optimization steps in each sample. The AlignNet model was trained using a vgg13 U-Net backbone with 3 coarse-to-fine levels. The input to the encoders for models trained with OSM data consisted of 3 channels of satellite data in rgb, combined with one-hot encoded information from the OpenStreetMap data for the node, line, and area channels. The satellite data and each openstreetmap channel went through a

$1 \times 1$ conv channel over 16 channels each, after which they were combined using a $3 \times 3$ conv channel over 64 channels. The model started with either the *center* or *lon - 1* starting positions. Furthermore, the L2 loss was directly optimized at the start as a warm up, but the influence of this loss decreased linearly to 0 at the end of the first epoch.

*4.2. Results*

A comparison of the resulting metrics on the KITTI dataset is shown in Table 2. We show the results on the cross-area test set, which consists of areas that are not part of the training set [13], and compare the previous state of the art with our best-performing model. Our model has the best latitude recall rate within 1 m and the best azimuth recall rate within 3 and 5 degrees. OrienterNet is the most capable model, obtaining better lateral and longitudinal results overall, but is unable to incorpate a rotation prior, resulting in a lower angular recall rate [12]. Overall, we find the results of our methods to be competitive with the current state of the art, showing the LM optimization algorithm to be a promising direction for small-scale localization problems.

**Table 2.** Recall rates on KITTI Cross-Area dataset. Best , second-best , and third-best results are marked. (a) Base model with both satellite and OSM data with the *lon - 1* starting position. The model is compared to the same model but with a certain modalities removed. When removing the lines modality, the results greatly suffer, while only keeping the satellite and line modalities keeps the accuracy. (b) New models trained to ablate over satellite input, OSM input and starting position. Certain input modalities can improve lateral and azimuth recall rates, but fall short on average.

| Name | Sat | OSM | Cross-Area | | | | | | | | |
| --- | --- | --- | --- | --- | --- | --- | --- | --- | --- | --- | --- |
| | | | Lateral | | | Longitudinal | | | Azimuth | | |
| | | | d = 1 | d = 3 | d = 5 | d = 1 | d = 3 | d = 5 | $\theta = 1$ | $\theta = 3$ | $\theta = 5$ |
| LM [13] | ✓ | ✗ | 27.82 | 59.79 | 72.89 | 5.75 | 16.36 | 26.48 | 18.42 | 49.72 | 71.00 |
| SliceMatch [11] | ✓ | ✗ | 32.43 | 78.98 | 86.44 | 8.30 | 24.48 | 35.57 | 46.82 | 46.82 | 46.82 |
| CCVPE [16] | ✓ | ✗ | 44.06 | 81.72 | 90.23 | 23.08 | 52.85 | 64.31 | 57.72 | 92.34 | 96.19 |
| OrienterNet (a) [12] | ✗ | ✓ | 51.26 | 84.77 | 91.81 | 22.39 | 46.79 | 57.81 | 20.41 | 52.24 | 73.53 |
| AlignNet (b) | ✓ | ✓ | 56.62 | 78.32 | 81.62 | 22.42 | 47.31 | 57.61 | 57.57 | 92.54 | 98.97 |

*4.3. Analysis of Model*

By taking a closer look of the behavior of the model and the features after the encoders, we can gain insights into the theoretical benefit of our model. The goal of this is to provide evidence of the possible superiority of this model compared to others. We first look at the qualitative results of the model iteration process and of our efforts to improve its ability to make estimations over iterations.To further show the behavior of the model over various iterations, we plotted the recall rate against the number of iteration steps for the different positional variables. We show that not all variables improve with increasing iterations, as would be expected of the model.

We first looked at the encoded features with an emphasis on the slight changes in the positional variables from the ground truth to see if the features encoded can show human-readable features such as line markings. We were, however, not able to generate useful insights using this methods as the deep encoded features were too far removed from human-readable features. We then also looked at the behavior of the Levenberg–Marguardt optimization algorithm by plotting the delta position for a range of each positional variable to show when the LM algorithm works well and when it does not work well. This proved to provide useful insight, especially regarding which starting position to use and especially why certain starting position strategies proved not to work to generate good results. Furthermore, we illustrated the final behavior of the model using heatmaps to show the inability of the model to find the correct position for various data points.

Model Behavior over Iterations

We looked at the behavior of the best-performing model, which utilized the OSM data and started the longitudinal variable at its minimal possible value. Figure 4 shows the success and failure cases of the model. As the loss function utilizes all intermediate positional losses, the model tries to jump close to the ground-truth position as fast as possible. To quantify how much the iterations help, we plotted the recall rate over all 15 steps in Figure 5. As shown, the recall rates of the lateral and longitudinal positional variables improved with increasing iterations. When looking at just the rotational variable, the model shows different behavior. The model, in fact, aims to obtain the correct angle very quickly, before even being close to the correct position. When the model is closer to the correct position, the expected behavior is that the model is better at obtaining the correct angle; however, the opposite is true. Even experiments which only added the azimuth loss for the final iteration did not show that the model improved the azimuth recall rate over the steps it took. This points to the model not optimally utilizing the ground-level image and instead using the knowledge of the camera mostly being aligned with the road direction to quickly obtain a good estimation.

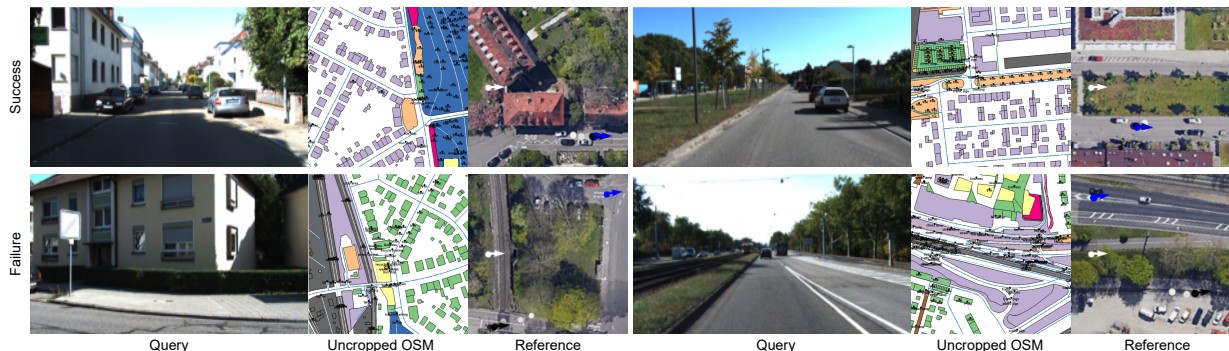

**Figure 4. Qualitative examples.** Two success and two failure cases for the KITTI cross-area dataset using our best-performing model. The model obtains the query and reference OSM and satellite data. Here, the OSM data are uncropped and unrotated to show the data available before the preprocessing steps. The white arrow on the reference satellite image shows the starting pose; the black arrow shows the ending pose, and the blue arrow shows the ground-truth pose. The dots show intermediate steps going from white to black over the steps.

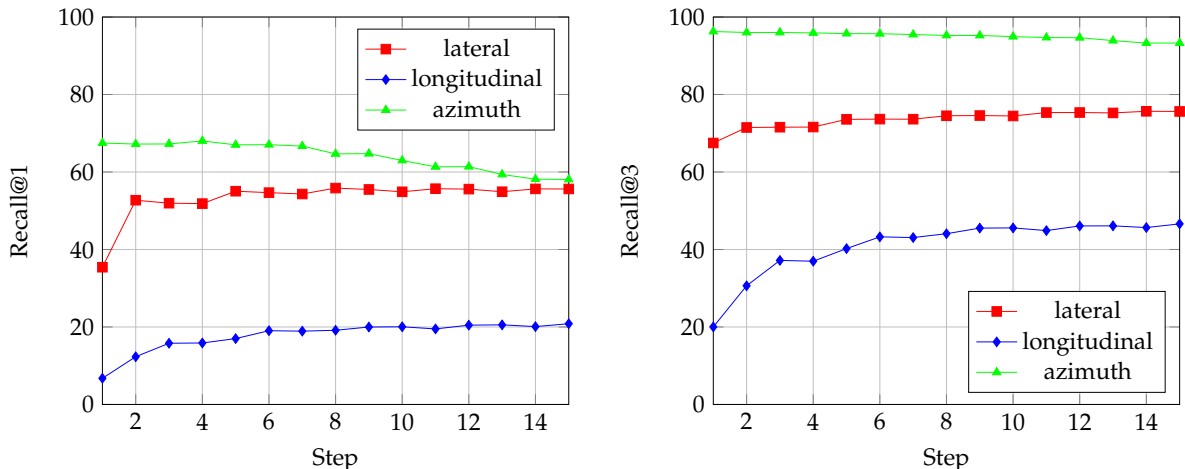

**Figure 5.** Percent of samples in the cross-area test set within 1 m/degree for the left graph and 3 m/degree for the right graph over the steps. The best-performing model was used, which utilized OSM data and started at the minimum longitudinal value. In both graphs, the lateral and longitudinal recall rates increased over the steps, while the azimuth recall rate slightly decreased.

*4.4. Ablation Studies*

In this section, we examine the different ablation studies we conducted. We ensure that the model makes use of the ground-level image for the final position estimation and look at the effectiveness of the satellite data, OSM data, and starting position. The goal of this is to provide an argument for the effectiveness of each part of the model, which we will also show with a contribution ablation study comparing the original LM model with our AlignNet.

### 4.4.1. Importance of Ground-Level Data for Alignment

If the model simply made a guess based on just the top-level image, it would invalidate the purpose of the model. To validate the use of the ground-level images, in Table 3, we compare the results of the images with and without the correct ground-level image. We show that for all but one category, the original ground-level image indeed shows better results. On the other hand, the results where the ground-level images are switched are still competitive and do not show large degradations. This shows that, while the model does seem to use the ground-level image to some extent, a large part of the good results are due to a good prior understanding of the possible places the image could have been taken. A reason for this could be the predictable nature of the dataset.

**Table 3. Ablation using different ground-level images.** The original model uses both satellite and OSM data and is trained for 26 epochs with the original ground-level image. The switched model uses another random ground-level image from the test dataset instead. The best results are shown in **bold**.

| Name | Cross-Area | | | | | | | | |
|---|---|---|---|---|---|---|---|---|---|
| | **Lateral** | | | **Longitudinal** | | | **Azimuth** | | |
| | d = 1 | d = 3 | d = 5 | d = 1 | d = 3 | d = 5 | $\theta = 1$ | $\theta = 3$ | $\theta = 5$ |
| original | **59.96** | **83.81** | 87.24 | **22.25** | **48.28** | **58.13** | **60.94** | **95.90** | **99.48** |
| switched | 59.39 | 83.78 | **87.32** | 19.50 | 45.83 | 57.52 | 57.49 | 94.93 | 99.35 |

### 4.4.2. OSM Ablation

We can see that the OSM data greatly help to improve the results, especially for the positional recall rates. The starting strategy has a large influence on the final result, with the model using OSM and with *lon - 1* achieving the best results for the AlignNet models (Table 4). We show that the right strategy allows the model to obtain competitive results compared to the state of the art.

### 4.4.3. Contribution Ablation

Finally, to show the different steps taken to go from the model introduced by Shi et al. to a model competitive with the recent successes seen in other papers, we trained different models to show the benefits of certain contributions, which are outlined in Table 5. The original LM model was trained with just 2 epochs, but we found the evaluation metrics steadily increased when training the model for more epochs. This increase in performance was consistent over the training, evaluation, and test sets. Increasing the U-Net model's size improved the metrics when looking at all positional variables together, but interestingly enough, this also resulted in a decrease in the lateral recall rate. Adding OpenStreetMap data mostly recovered the lateral recall rate, but so far, all the models struggle on the longitudinal metrics. Finally, by starting the model from the minimum longitudinal value, we ended up with a model with all-around good performance.

**Table 4. Recall rates on KITTI Cross-Area dataset.** Best , second-best , and third-best results are marked. (a,b) Base model with both satellite and OSM data with the *lon - 1* starting position. The model is compared to the same model but with a certain modalities removed. When removing the lines modality, the results greatly suffer, while only keeping the satellite and line modalities keeps the accuracy.

| Name | Sat | OSM | Start | Cross-Area | | | | | | | | |
| --- | --- | --- | --- | --- | --- | --- | --- | --- | --- | --- | --- | --- |
| | | | | Lateral | | | Longitudinal | | | Azimuth | | |
| | | | | d = 1 | d = 3 | d = 5 | d = 1 | d = 3 | d = 5 | $\theta = 1$ | $\theta = 3$ | $\theta = 5$ |
| | ✓ | ✓ | *lon - 1* | 56.62 | 78.32 | 81.62 | 22.42 | 47.31 | 57.61 | 57.57 | 92.54 | 98.97 |
| | ✓ | only nodes | *lon - 1* | 17.50 | 37.39 | 46.43 | 9.10 | 24.26 | 34.83 | 27.41 | 71.88 | 96.67 |
| Ours (a) | ✓ | only lines | *lon - 1* | 58.71 | 79.66 | 83.49 | 19.89 | 45.32 | 55.33 | 57.96 | 92.85 | 99.00 |
| | ✓ | only areas | *lon - 1* | 15.79 | 35.71 | 45.17 | 9.25 | 24.53 | 35.04 | 26.80 | 72.57 | 96.83 |
| | ✓ | ✗ | *lon - 1* | 17.89 | 38.12 | 46.80 | 8.91 | 24.16 | 34.78 | 27.31 | 72.02 | 96.65 |
| | ✓ | ✗ | center | 42.96 | 73.97 | 81.20 | 12.56 | 32.47 | 43.07 | 58.29 | 96.90 | 99.67 |
| | ✓ | ✗ | *lon - 1* | 40.71 | 69.54 | 76.08 | 5.30 | 15.08 | 23.15 | 58.13 | 92.95 | 97.71 |
| AlignNet (b) | ✓ | ✓ | center | 50.13 | 78.61 | 84.01 | 7.12 | 20.21 | 31.66 | 49.70 | 78.71 | 87.58 |
| | ✗ | ✓ | center | 52.82 | 80.52 | 85.11 | 10.30 | 28.48 | 42.35 | 44.84 | 75.44 | 85.93 |
| | ✗ | ✓ | *lon - 1* | 58.82 | 81.08 | 84.46 | 14.88 | 36.77 | 49.18 | 49.87 | 77.26 | 86.38 |

**Table 5.** Ablation of contributions compared to original LM model. Best , second-best , and third-best results are marked. The original model was trained for just 2 epochs and thus did not achieve convergence. Increasing the amount of training the model underwent and the size of the U-Net using a more conventional architecture improved the recall rate. Using OSM data and starting at the minimum longitudinal value improved the model to competitive accuracy.

| Name | Cross-Area | | | | | | | | |
| --- | --- | --- | --- | --- | --- | --- | --- | --- | --- |
| | Lateral | | | Longitudinal | | | Azimuth | | |
| | d = 1 | d = 3 | d = 5 | d = 1 | d = 3 | d = 5 | $\theta = 1$ | $\theta = 3$ | $\theta = 5$ |
| LM [13] | 27.82 | 59.79 | 72.89 | 5.75 | 16.36 | 26.48 | 18.42 | 49.72 | 71.00 |
| +train 21 epochs | 40.81 | 75.38 | 83.53 | 5.42 | 17.14 | 28.08 | 25.40 | 61.03 | 78.87 |
| +bigger U-Net | 41.08 | 71.69 | 79.65 | 11.58 | 30.05 | 44.59 | 64.27 | 98.46 | 99.91 |
| +OSM | 50.13 | 78.61 | 84.01 | 7.12 | 20.21 | 31.66 | 49.70 | 78.71 | 87.58 |
| +start at *lon - 1* | 56.62 | 78.32 | 81.62 | 22.42 | 47.31 | 57.61 | 57.57 | 92.54 | 98.97 |

## 5. Discussion and Conclusions

In this paper, we introduced AlignNet, a cross-view outdoor localization method for augmented reality that is based on fusing map and satellite data. The model is an extension of the previous LM model and improves it in such a way as to be competitive on the task of small-scale localization on the KITTI dataset. For this work, we introduced the fusion of satellite and OSM data and changed the architecture and training procedure to beat the state of the art's lateral recall rate within 1 m and azimuth recall rate at 3 and 5 degrees. We analyzed the behavior of our best-performing model and conducted an ablation study to show the benefits and shortcomings of the model.

When analyzing our best-performing model, we found that the model is able to align the positional variables over its steps up to a certain point but still relies heavily on a good initial guess at the first step the model makes. This suggests that the model in its current form does not yet take full advantage of its iterative nature but does point towards sample-wise optimization as a viable path of research going forward. Furthermore, we showed that OSM data is highly beneficial in the task of small-scale visual localization. For the lateral positional variable, we show that the recall rate within 1 m receives a boost of up to 18%.

In the end, we have shown that our sample-wise optimization strategy is a viable alternative to other methods and have shown a promising direction for future research in the field of visual localization.

**Author Contributions:** Conceptualization, M.R.O. and S.d.H.; Methodology, M.R.O. and S.d.H.; Investigation, R.E., S.d.H. and D.D.; Writing—original draft, R.E.; Writing—review & editing, D.D.; Supervision, M.R.O., S.d.H. and D.D. All authors have read and agreed to the published version of the manuscript.

**Funding:** This research received no external funding.

**Informed Consent Statement:** Not applicable.

**Data Availability Statement:** Data from this study can be found at Planet Dump repository https://planet.osm.org (accessed on 10 June 2023).

**Conflicts of Interest:** The authors declare no conflict of interest.

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
