# Peer review of "Cross-View Outdoor Localization in Augmented Reality by Fusing Map and Satellite Data"

_applsci, doi:10.3390/app132011215_

Round 1

Reviewer 1 Report

In this manuscript, authors proposed a deep learning based visual localization method (AlignNet) by matching ground-level images with top-down imagery. AlignNet combines three modules: a two-branch U-Net, geometric projection and LM optimization to estimate the longitude, latitude and angle of a ground-level image. Moreover, the dataset includes OpenStreetMap (OSM) data and satellite data which also improve the performance of the localization. By The manuscript got competitive results compared with SOTA methods on the KITTI cross-view dataset. This article presents an interesting method, but there are some points that need improvement/clarity before it can be accepted.

1. The description of spatial resolution (cm per pixel) in the dataset is inconsistent. For satellite images (1280 pixels by 1280 pixels ) with a range of 100 meters, the spatial resolution is less than 10 cm. However, the following description in the manuscript mentions that the spatial resolution of OSM imagery is 50 cm. (line 138)

2. How does the dataset used in the manuscript determine the spatial resolution and latitude-longitude range of the images after rotation? (line 126: at most 10 degrees)

3. In the manuscript, it is mentioned that setting the longitude to -1 during initialization can improve the performance of localization. Would setting the latitude to -1 also lead to a similar improvement?

4. There are some typos in the manuscript. For example, line 34,35: matching matching, line 150: [? ], line 168. Please carefully review the manuscript and make corrections.

Author Response

Thank you very much for taking the time to review the manuscript. We include responses to the comments and the corresponding changes to the paper.

1. This is indeed a mistake. Line 121 should be 250m by 250 area. This is changed in the new version.

2. For the training set the rotation range is between -10 and +10 degrees. The ranges are pre-determined such that for every run the same random degree is used for each image for fair comparision. We have changed the paragraph to make the preprocessing step a bit more clear, from line 125 to 130. 
Before: The input data, before going into the encoders, is first randomly rotated and translated, by at most 10 degrees and 40 meters in any direction. Afterwards, an area of 40m by 40m is cut out as input to the model. For the OSM data the same preprocessing steps are taken to ensure the satellite and OSM data are exactly aligned.
After: Before the input data is put into the encoders, the translation and rotation noise is added to the image. The center 100m by 100m area, corresponding to an image of 512 by 512 pixels, is cut out. This area is larger than the maximum translation noise to ensure the model work properly even at the border of the translation noise border. For the OSM data the same preprocessing steps are taken to ensure the satellite and OSM data are exactly aligned.

3. The model starts with a angle of 0 degrees, which corresponds to looking towards the right side of the image, which we call the longitudinal direction. The model is unable to see towards the negative longitudinal direction, making it important to start at the longitudinal -1 position. The model however is able to look towards both the positive and negative lateral direction, which makes starting in the middle optimal. We did look at randomizing the starting lateral position to make the model more generalizable but that did not show improvements.

4. We did a thorough check over the manuscript to fix these issues, including the ones raised in this point.  

Reviewer 2 Report

This manuscript studied a new algorithm to match the ground-level image with overhead imagery. Specifically, the author introduced the Alignnet model, which achieved a fourfold improvement in recall rate on a visual position dataset. The manuscript's structure is clear, and the approach is interesting. Therefore, I recommend its publication in the Applied Science journal. I just proposed a few minor issues:

1. In the third box of Figure 3, the arrow above Yl,i is slightly tilted, the authors should make it vertical.

2. Line 150: There is an issue with the reference citation, the authors should advise it.

Author Response

Thank you very much for taking the time to review the manuscript. We include responses to the comments and the corresponding changes to the paper.

1 & 2. We agree with the points made and improved it in the updated version. We did a thorough check over the manuscript to fix issues like this including the two issues raised here.

Reviewer 3 Report

Interesting work. For details, please check the attached file.

Author Response

Thank you very much for taking the time to review the manuscript. We include responses to the comments and the corresponding changes to the paper.

1. Thanks for the many spotted spelling mistakes. We have improved upon the spelling in the updated version including all the examples given.

2. We have changed those two lines to improve the clarity of the manuscript. The second line is changed from Given a rotation given by rotation matrix R and translation given by t we calculate to Given a rotation given by matrix R and translation by vector t we calculate

3. Thanks for these grammar mistakes. We have fixed many grammatical mistakes in the updated version including all the examples given.

4. When extracting features from OSM, do you classify all of them? For example in the case of nodes which field do you use to make classes? Do you apply some data cleaning?

We manually make rules to map nodes/lines/areas with certain tags to certain classes. When a node does not contain any useful tags we map them to null and do not use it as input to the model. Using this process, we have created many rules such that it is able to extract useful OpenStreetMap input for areas outside of the dataset area as well without manually adding extra rules. As this process of making new rules is not automatic, and as we did not compare different data extraction strategies we did not go in depth into this. That said, it would be better to provide a little more information than what was written previousily, so we changed the description to: 
Following work by Sarlin et. al. we, instead of directly taking a screenshot-like image of OpenStreetMap, extract node, line and area (closed-loop line objects) objects. Based on the tags included in these objects, we construct rules to map useful objects to different classes.

You should highlight more clearly how your work is related to AR. Why do you need visual
localization in AR, where the screen of our device shows the real word and some pictures are superimposed?

This type of visual localization is especially useful for navigation in AR. As such, we added this example to the introduction on line 23-23:
for the purpose of applications such as augmented reality. ->
for the purpose of augmented reality applications where the precise location is important such as AR navigation.

Could you explain what ‘works well’ means in line 193?

We used works well as little experimentation for different values resulted in lambda=0.1 proving to be a good choice. As we did not do extensive experimentation for this value, we changed the line to refer to prior work instead as they use this specific value as well:
We found that the value of lambda=0.1 works well for the given task. -> We use the same value of lambda=0.1 as prior work
